# Opening a Novel Biosynthetic Pathway to Dihydroxyacetone and Glycerol in *Escherichia coli* Mutants through Expression of a Gene Variant (*fsaA*^A129S^) for Fructose 6-Phosphate Aldolase [note 1]

**DOI:** 10.3390/ijms21249625

**Published:** 2020-12-17

**Authors:** Emma Guitart Font, Georg A. Sprenger

**Affiliations:** Institute of Microbiology, University of Stuttgart, Allmandring 31, D-70569 Stuttgart, Germany; e.guitartfont@gmail.com

**Keywords:** *Escherichia coli*, phosphofructokinase, *pfkAB zwf* triple deletion, fructose 6-phosphate aldolase, point mutation, new-to-nature pathway, metabolic bypass, dihydroxyacetone, glycerol dehydrogenase, glycerol formation

## Abstract

Phosphofructokinase (PFK) plays a pivotal role in glycolysis. By deletion of the genes *pfkA*, *pfkB* (encoding the two PFK isoenzymes), and *zwf* (glucose 6-phosphate dehydrogenase) in *Escherichia coli* K-12, a mutant strain (GL3) with a complete block in glucose catabolism was created. Introduction of plasmid-borne copies of the *fsaA* wild type gene (encoding *E. coli* fructose 6-phosphate aldolase, FSAA) did not allow a bypass by splitting fructose 6-phosphate (F6P) into dihydroxyacetone (DHA) and glyceraldehyde 3-phosphate (G3P). Although FSAA enzyme activity was detected, growth on glucose was not reestablished. A mutant allele encoding for FSAA with an amino acid exchange (Ala129Ser) which showed increased catalytic efficiency for F6P, allowed growth on glucose with a µ of about 0.12 h^−1^. A GL3 derivative with a chromosomally integrated copy of *fsaA*^A129S^ (GL4) grew with 0.05 h^−1^ on glucose. A mutant strain from GL4 where *dhaKLM* genes were deleted (GL5) excreted DHA. By deletion of the gene *glpK* (glycerol kinase) and overexpression of *gldA* (of glycerol dehydrogenase), a strain (GL7) was created which showed glycerol formation (21.8 mM; yield approximately 70% of the theoretically maximal value) as main end product when grown on glucose. A new-to-nature pathway from glucose to glycerol was created.

## 1. Introduction

Glucose metabolism in most bacteria starts via its uptake and phosphorylation (either by a PTS or via glucokinase) to glucose 6-phosphate (G6P) (for reviews: [1,2]). G6P is subsequently broken down to pyruvate mainly via the Embden–Meyerhof–Parnas pathway (EMP), and in part by the hexose monophosphate shunt (HMP), or the Entner–Doudoroff (ED) pathway [3]. In the EMP, G6P is isomerized by phosphoglucose isomerase (PGI) to fructose 6-phosphate (F6P). The first committed step in EMP glycolysis is catalyzed by phosphofructokinase (fructose 6-phosphate 1-kinase, PFK) which phosphorylates F6P to fructose 1,6-bisphosphate (F1,6BP) in a practically irreversible reaction [4,5]. PFK thus plays a pivotal role in the regulation of and flux through the EMP pathway of glycolysis. Furthermore, the oxidative pentose phosphate pathway (PPP) also depends on PFK activity. The model microorganism, *Escherichia coli* (*E. coli*) K-12 has two genes encoding PFK activity (*pfkA*, *pfkB*) [6,7]; the encoded isozymes are quite different as they display distinct kinetic features and they are differently regulated [4,8].

Mutations which led to the loss of the major activity (PFKA, alternatively abbreviated in the literature as PFK-1) already yielded growth defects on carbon sources such as glucose or hexitols (mannitol, glucitol/sorbitol) [6,7]. *E. coli pfkA* mutants gave rise to suppressor mutations, (e.g., *pfkB*1) with elevated enzyme activity of PFK-2 (PFKB) [6,7,9]. Double deletions (Δ*pfkA*, Δ*pfkB*) are devoid of PFK activity and showed severe growth defects on several sugars such as glucose [10,11,12,13]. Growth of *pfk* mutants on fructose, however, is less affected as this hexose mainly enters via the fructose PTS (EII^Fru^) which phosphorylates it to fructose 1-phosphate which is further converted by a fructose 1-phosphate 6-kinase [14].

In *pfk* mutants, G6P may be channeled via glucose 6-phosphate dehydrogenase (ZWF) to the oxidative pentose phosphate pathway (PPP) and/or to the Entner–Doudoroff (ED) pathway of glycolysis [6,7]. From the PPP, glyceraldehyde 3-phosphate (G3P) could enter the lower glycolytic trunk [15]. A deletion of the *zwf* gene would block both the entry to the oxidative PPP and to the ED pathway. Recently, three groups have reported the construction of *E. coli* triple deletion strains (Δ*pfkA*, Δ*pfkB*, Δ*zwf*) but did not describe growth behavior on glucose [16,17,18]. A Δ*zwf* Δ*pfkB pfkA*::114-*pfkA* (DAS + 4) “PFK-knockdown” mutant of strain MG1655 has been used for redirecting flux from G6P to inositol formation [19]. Such a triple-negative mutant would be expected to be devoid of growth on glucose and several other C sources (Figure 1).

Our group has discovered two genes (*fsaA*, *fsaB*) in the *E. coli* K-12 chromosome which both encode fructose 6-phosphate aldolase (FSA) activity (forming dihydroxyacetone (DHA) and G3P from F6P in a reversible manner) when expressed from recombinant vectors [20]. Both genes, however, are expressed at very low rates from their chromosomal locations ([21,22], and our own results); estimations for a single *E. coli* cell (after growth in minimal media) gave values in the range of 100 protein molecules of FSAA (gene product of *fsaA* gene) or even less for FSAB [22]. The physiological roles (in vivo) of the two genes therefore remain enigmatic [23]. Recombinant FSAA as an in vitro aldolase biocatalyst, however, has been the subject of many investigations in chemo-enzymatic syntheses as FSAA displays an unusual broad substrate spectrum encompassing not only DHA but also other donor compounds such as hydroxyacetone (acetol) or glycolaldehyde [24,25,26,27].

For utilization of FSA activity in vivo, several groups have successfully utilized the cloned *fsaA* gene for metabolic pathway engineering of *E. coli*. By doing so, growth on DHA [28], growth on glycerol via DHA [29], and a new pathway for the synthesis of glycolic acid [30] were accomplished. As well, a novel road to the formation of 1-deoxyxylulose from hydroxyacetone was opened which allows for a novel access to the methylerythritolphosphate biosynthesis pathway of isoprenoids in *E. coli* [31].

From theoretical considerations, the FSA retro-aldol reaction has already been introduced as a constituent in several genome scale metabolic models for *E. coli* [32,33,34] but, to the best of our knowledge, experimental proof for such a role in cleavage of F6P has not been provided so far. We wanted to find out whether the wt *fsaA* gene (recombinantly expressed either from a plasmid vector or after insertion into the chromosome, in both cases under the control of the IPTG-inducible promoter P_tac_) would reinstall growth on glucose in a triple mutant of *E. coli* K-12 which lacks PFK and ZWF activities. We show here that, while protein formation and sufficient enzyme activity was detectable in cell-free extracts, no growth was observed with the recombinant wild type gene. Albeit, a mutant allele (*fsaA*^A129S^) [24,27,35] conferred a glucose-positive phenotype, both when expressed from a plasmid or as single copy in the chromosome.

DHA is an ingredient in many self-tanning compounds and glycerol is a carbon source which has received much attention as it can be used for production of many added-value compounds (for references see [36]). In contrast to other microorganisms such as yeast [37], cells of *E. coli* are not known to form and excrete glycerol when growing on glucose [38,39]. By introduction of genes from yeast which encode enzymes for the conversion of DHAP to glycerol, however, *E. coli* glycerol producers had been constructed [40,41,42,43]; eventually this has led to recombinant *E. coli* cells forming 1,3-propanediol from glucose via the intermediate glycerol. 1,3-propanediol is now used as monomer for the industrial production of the biobased polymer polytrimethylene terephthalate (PTT) [36,41]. As we observed transient formation of DHA from glucose with the newly developed strains described here, we decided to aim for a novel pathway to DHA and glycerol from glucose in *E. coli* mutants. This pathway is completely independent of heterologous genes.

## 2. Results

### 2.1. Construction and Characterization of an E. coli Triple Mutant Strain Deficient in Phosphofructokinase (Genes pfkA, pfkB) and Glucose 6-Phosphate Dehydrogenase (zwf)

*E. coli* K-12 W3110 (wild-type) strain LJ110 [44] was chosen as the parent strain. In order to block glycolysis at the step of F6P and also to prevent a glycolytic bypass through the HMP, we successively deleted the genes *zwf*, *pfkB*, and *pfkA* encoding activities of glucose 6-phosphate dehydrogenase and both phosphofructokinases [45,46], respectively, by λred recombineering [47]. For details of strain construction see Section 4 and Table 1 for genotypes of strains.

The growth phenotype of the *E. coli* LJ110 wild type (wt) and the triple mutant GL3 (Δ*pfkA*::FRT Δ*pfkB*::FRT Δ*zwf*::FRT), was then compared on various media (Luria Bertani medium, LB, minimal medium (MM) with different single carbon sources; Table 2). We chose sugars and sugar alcohols which enter glycolysis above or below F6P. Growth on minimal agar plates was examined over a course of 2 days at 37 °C. As can be seen from Table 2, the wt strain LJ110 grew well on all tested carbon sources whereas the triple mutant strain GL3 showed several aberrations. While growth on LB was slightly affected, no growth was detected on MM agar plates or in shake flasks with the sole carbon sources D-glucose, mannitol, or sorbitol (all PTS substrates), and neither with lactose, D-galactose, maltose, or D-xylose. Growth on D-fructose was slower and smaller colonies were formed on MM agar plates.

Thus, the combination of the three mutations (*pfkA*, *pfkB*, *zwf*) led to less growth on sugars entering glycolysis at the level of or above from F6P (see Figure 1). Gluconeogenic substrates as glycerol, D- or L-lactate, and succinate also yielded slower growth; so did D-gluconate and L-arabinose. This lack of growth might be due to intracellular accumulation of F6P or result from regulatory defects. A comparison of growth in liquid MM (shake flasks) of LJ110 wt strain and GL3 on glucose and fructose is given in Figure 2. Whereas wt strain LJ110 grew on glucose (µ 0.59 ± 0.01 h^−1^) and fructose (µ 0.55 ± 0.04 h^−1^) with similar growth rates, GL3 grew on fructose as sole carbon source with a µ of 0.26 ± 0.01 h^−1^ and no growth was observed in MM glucose over a range of 96 h. Although D-fructose, as a single C source, enabled GL3 to grow at a reduced growth rate, we decided to use D-fructose as a neutral C source in preculture media for growth experiments to avoid long lag phases in MM subcultures when using other C sources.

We hypothesized that—by prolonged incubation on MM agar plates with glucose as sole carbon source—mutant colonies of GL3 might appear due to activation of the inherent chromosomal genes *fsaA* and/or *fsaB* genes which could suppress the glucose-negative phenotype. However, even after a prolonged incubation of 3 weeks, no such suppressor mutants appeared. Growth on glucose, however, could be restored in strain GL3 when it was transformed with a plasmid carrying wild type *pfkA* gene (pJNTN-m-L-*pfkA*; Table 3) (data not shown). We take this as evidence that the growth deficiency of strain GL3 mainly lies in its inability to degrade F6P or glucose 6-phosphate (G6P) by the EMP route (due to blockage of both PFKs) or by the hexose monophosphate (HMP) route (lack of glucose 6-phosphate dehydrogenase).

### 2.2. Expression of a Mutant Gene (fsaA^A129S^) Restores Growth of GL3 on Glucose

Recombinant FSAA is known to perform its eponymous reaction, F6P cleavage (retro-aldol reaction), in vitro [20]. We wanted to find out whether a plasmid-based expression of the *E. coli fsaA* gene might open up a bypass reaction for glycolysis by cleavage of F6P to DHA and G3P in vivo. Therefore, the genes encoding the wild type FSAA or a mutant gene encoding the variant FSAA A129S, which has an improved catalytic efficiency towards F6P [24,27,35] were tested. These genes had been cloned earlier into the pJF119EH expression plasmid which contains an ampicillin resistance marker for selection (see Table 3) and a *lacI* gene for repression in the absence of inducer. Both recombinant *fsa* genes are under the control of an IPTG-inducible P_tac_ promoter [51,52]. Both plasmids were introduced in GL3; as a control, the empty vector (pJF119EH [51]) was used. Expression of *fsaA* gene was examined in strain GL3 after growth on MM fructose and was verified by SDS-PAGE and enzyme analyses. As can be seen (Appendix A), cell-free extracts (cfe; soluble protein fraction) of GL3/pJF119*fsaA* after growth on MM with 28 mM fructose and 100 µM IPTG showed a prominent additional protein band at the size of approx. 24 kDa which is typical for FSAA [20]. This band was missing in cfe of wt strain LJ110. FSAA is known to be stable for a prolonged time at elevated temperatures (up to 75 °C) in vitro and retains most of its enzyme activity after such a temperature treatment [20]. Heat treatment at 75 °C for 20 min indeed led to an enrichment of the ≈24 kDa protein band in the supernatant of this strain (see Appendix A). FSAA enzyme activity (cleavage of F6P to DHA and G3P) of the heat-treated extracts of GL3/pJF119*fsaA* was determined to be in the range of 0.7 ± 0.0 U/mg of cell-free protein. Knowing that the plasmid-borne wild type gene *fsaA* was sufficiently expressed and active in *E. coli* strain GL3 (yielding soluble protein and active FSA enzyme in vitro), we expected that growth on MM glucose should be restored by expression of the recombinant gene copy. As can be seen from Figure 3, however, no growth on glucose occurred for 96 h with strains GL3 and GL3/pJF119EH (controls), or with GL3/pJF119*fsaA*.

From Figure 3 it also becomes evident that GL3 carrying the mutant gene *fsaA*^A129S^ was able to grow on glucose and without a noticeable lag-phase. The doubling time was determined at about 6 h (359 ± 52 min) corresponding to a growth rate (in the early exponential phase) of 0.12 ± 0.01 h^−1^.

To the best of our knowledge, this is the first report that expression of an *fsa* gene restores growth on glucose in a mutant background which presumably accumulates F6P by following the eponymous function of its encoded protein. FSAA A129S activity from heat-treated cfe (after growth in MM with glucose and 100 µM IPTG) was in the range of 4.9 ± 1.7 U/mg of protein.

The feature that plasmid-borne *fsaA*^A129S^ confers a glucose-positive growth phenotype could also be seen on MM agar plates with glucose as sole carbon source (in the presence of IPTG). When cells of GL3 were transformed with plasmid DNA of pJF119*fsaA*^A129S^, about equal transformation efficiencies (colony forming units per µg of DNA) were found on LB with 100 µg/mL ampicillin agar plates compared to MM glucose + IPTG agar plates (data not shown).

### 2.3. Transient Formation of Dihydroxyacetone from Glucose

While the in vivo presence of FSAA A129S (as observed by SDS-PAGE and enzyme activity in heat-treated extracts) and the restoration of growth in GL3 nicely fit with the assumption of a bypass reaction at the metabolic block at F6P (see Figure 4), further evidence for the in vivo functionality of this enzyme is necessary.

Cleavage of F6P should liberate DHA and G3P in stoichiometric amounts. G3P could be directly further metabolized in the lower trunk of glycolysis to provide energy and reducing equivalents as well as PEP for glucose uptake via the PTS; furthermore, through action of transketolase on F6P and G3P the PPP could be supplied. The fate of DHA liberated intracellularly in *E. coli* would be less clear as DHA is not considered as usual intermediate of central metabolic pathways [55]. Therefore, we looked for the appearance of DHA in culture supernatants of GL3/pJF119*fsaA*^A129S^ during growth on glucose (≈30 mM initial concentration). Glucose consumption and DHA appearance was measured by HPLC, results are shown in Figure 5. As can be seen, with the onset of fast glucose consumption (after about 12 h of incubation) DHA appeared in the supernatant and reached a maximal concentration of approx. 12 (± 4) mM after 60 h of incubation. Thereafter, DHA successively disappeared from the medium and was no longer detected at 96 h, while glucose was already completely consumed at about 80 h.

This observation could be explained by DHA reuptake from the medium and metabolism, putatively by the PEP-dependent DHAKLM system [55,56,57,58]. We take the appearance of DHA (in millimolar concentrations) as further evidence that the activity of FSAA A129S is indeed necessary to restore growth of the triple mutant GL3 on glucose. Our efforts to improve the formation of DHA are shown below. Figure 4 summarizes the putative bypass pathway at F6P which allows growth on glucose even if PFK is missing.

### 2.4. A Single Chromosomal Copy of fsaA^A129S^ Is Sufficient for Growth of the Triple Mutant

As the mutant gene *fsaA*^A129S^ had been expressed from a multicopy plasmid vector [51], we wanted to know whether a single copy of the mutant gene in the chromosome of GL3 would suffice to allow growth on glucose. Therefore, by insertion of the *fsaA*^A129S^ gene under the control of P_tac_ at the *lacZ* locus of GL3, strain GL4 was constructed; this strain is plasmid-free and, thus, allows study in the absence of antibiotic selection. GL4 grew indeed on MM glucose though with a lower growth rate (µ approx. 0.05 h^−1^) than GL3/pJF119*fsaA*^A129S^ as can be seen in Figure 6. During growth, only minor DHA concentrations were detected (0.5 mM after 30 h of incubation; see Table 4). FSA activity in heat-treated cfe was approx. 0.1 U/mg of protein.

As we aimed for a better DHA formation, we were interested to see how genes of the glycerol and DHA metabolism were expressed in strain GL4; as well we looked for gene activities such as *sgrS* which we expected to be changed (see below). Selected transcript levels were therefore compared to the ones of wt LJ110 by qPCR. As can be seen from Appendix A, qPCR transcript analysis showed that, as expected, no transcripts for *zwf, pfkA, pfkB*, *lacZ* could be detected in RNA preparations from strain GL4 as these genes had been deleted through strain construction. Gene expression of *fsaA* was barely seen in LJ110, but was clearly observed in GL4. Note that we cannot discern by the chosen method between transcript from the *fsaA*^A129S^ allele at the *lacZ* locus and the inherent *fsaA* gene at its natural locus. Genes for glycerol metabolism, *gldA* and *glpK*, were expressed at very low levels in both strains; *dhaK* was expressed at clearly higher rates in GL4 than in LJ110. Therefore, we think that in strain GL4, DHA is phosphorylated to DHAP by the PEP-dependent DHAKLM and thus can be further metabolized.

We also looked for transcript amounts of *sgrS* which is an indicator for G6P accumulation and a regulator of glucose PTS [59]. While transcripts were below the detection threshold in LJ110, *sgrS* was clearly expressed in GL4. Moreover, *ptsG* was expressed in GL4 at a lower rate, presumably due to regulation by *sgrS*. The occurrence of *sgrS* RNA in GL4 could be interpreted as a sign of glucose phosphate stress which has an influence on glucose uptake in *E. coli* [59,60].

### 2.5. A Novel Pathway of Glycerol Formation in E. coli

As described in the introduction, cells of *E. coli* are not known to form and excrete glycerol when growing on glucose [38,39]. As we showed that GL3/pJF119*fsaA*^A129S^ (Figure 5), and to a lesser extent also GL4 (Table 4), excrete DHA during growth on glucose, we intended to stop the observed catabolism of DHA in order to favor DHA and glycerol formation from sugars. We therefore deleted the chromosomal *dhaKLM* genes in strain GL4 to yield strain GL5. The deletion in this case was performed by the novel CRISPR-Cas9 technology (see Table 1) [48].

GL5 grew slower on glucose (µ of 0.02 h^−1^) than strain GL4 (see Table 4) and only after a prolonged growth lag. GL5 formed DHA (maximal 3.1 mM) transiently but eventually consumed it again, presumably via the inherent *gldA* (glycerol dehydrogenase) and *glpK* (glycerol kinase) gene activities. The deletion of *dhaKLM* genes apparently had not yet abolished DHA consumption. Next, the *glpK* gene was knocked out in strain GL5 to prevent any glycerol consumption (strain GL6, see Table 1). Subsequently, *gldA* was cloned and integrated in the chromosomal *rbsK* locus (under the control of P_tac_) to come up with strain GL7 (see Table 1 for construction details).

Then we studied the growth behavior of strains GL6 and GL7 and their product spectra. When incubated on MM containing glucose and IPTG, GL6 required approx. 288 h to reach maximal optical density, while GL7 took about 240 h (see Table 4 and Appendix A). Heat-treated cfe of GL6 and GL7 contained FSA activities each of about 0.1 U/mg. Glycerol dehydrogenase (GLDA) activity in cfe of GL7 was approx. 0.4 (± 0.1 U/mg), whereas GLDA activity in GL6 was below detection level. In the supernatant of GL6, approx. 8.8 mM DHA were found, in GL7 approx. 3.5 mM (see Table 4). As a novel product, glycerol, was detected in GL6 with a concentration of 1.9 ± 0.2 mM, in GL7 this value increased about 11-fold to 21.8 (see Table 4). Both products (DHA and glycerol) remained in the supernatant until the end of the incubation. In the case of GL7, the sum of both C3 products is about 25 mM and thus about 84% of the theoretical maximal yield (based on C molarity of glucose, a C6 compound, which was consumed).

We were then interested to see whether strain GL7 could also utilize other sugars for glycerol production, especially those which are constituents of hemicellulose. Therefore, D-xylose, L-arabinose, and D-galactose were used as single C-sources (see Table 4). Initial pentose (C5 compounds) concentrations were set at approx. 33 mM to be comparable to the initial 28 mM of hexoses (165 mM C atom compared to 168 mM C atom). With D-galactose, GL7 needed only half the time to reach the maximal OD_600 nm_ than on glucose (120 h against 240 h). The pentoses D-xylose and L-arabinose (constituents of hemicellulose in plants) were also used (see Figure 7 and Table 4) for growth and glycerol was yielded as product.

After growth on D-galactose, approximately 18 mM glycerol was found which is about 4 mM less glycerol in comparison to growth on D-glucose. Glycerol concentrations were 13.3 mM for D-xylose and 13.9 mM for L-arabinose. In terms of atom molarity (expressed in the amounts of glycerol produced) this gives the order:

D-glucose > D-galactose > L-arabinose~D-xylose.

[Calculation 21.8 × 3 = 65.4 mM C from 90.6 mM C for D-glucose (≈72% theoretical maximal value), 17.9 × 3 = 53.7 mM C for 84.6 mM C of D-galactose (≈63%); for D-xylose: 39.9 mM C from 98.7 mM C (≈40%), and 41.7 from 105 mM C (≈40%) for L-arabinose].

Please note that the molar yield on glycerol/DHA is less than theoretically possible. For possible reasons see Section 3.

## 3. Discussion

In this work, by deletion of the genes for PFKA and PFKB (*pfkA, pfkB*) as well as of *zwf* (glucose 6-phosphate dehydrogenase), a glucose-negative mutant of *E. coli* K-12 was created. Very recently, other groups have reported the construction of such triple-negative strains, however, they were interested in different features and did not report the glucose phenotype [16,17,18]. Thus, we present here the broader characterization of a ∆*zwf* ∆*pfkB* ∆*pfkA* triple deletion strain (such as GL3) on various C sources. GL3 was found to be unable to grow on C sources which are taken up by the PEP-dependent sugar:phosphotransferase system (PTS) such as glucose, *N*-acetylglucosamine, mannose, mannitol, and others. These substrates have in common that they enter the EMP at F6P or G6P [6]. Although glucose in *E. coli* might be oxidized via a PQQ-dependent glucose dehydrogenase to gluconic acid and thereby could circumvent the coerced genetic blocks, this oxidation apparently did not occur in GL3, most likely because PQQ as cofactor cannot be formed by *E. coli* cells [61,62]. As well, GL3 showed severe impairments when growing on D-galactose, glucose 6-phosphate, glycerol, pentose sugars, or other C sources which have to pass through F6P. Degradation of sugars which enter glycolysis at the stage of DHAP such as L-fucose or L-rhamnose [63], however, was little affected in GL3 (data not shown); growth of GL3 on D-fructose was still possible, presumably because it can enter the EMP at the stage of F1,6BP [14].

There is ample evidence for microbial cells being able to cope with genetic perturbations of central metabolic pathways (including blocks in the EMP) by activation of inherent metabolic pathways [64,65]. In line with several genome scale metabolic models for *E. coli* which already had incorporated FSA as a constituent of central metabolism [32,33,34], we had expected that a block in PFK in *E. coli* might be overcome by activation of the dormant genes *fsaA* or *fsaB* as these genes encode a F6P aldolase activity [20]. However, even after prolonged incubation with glucose as sole C source, we did not detect GL3 mutants which could grow again on glucose. We therefore focused on the question whether recombinant expression of the gene for FSAA on a plasmid might create a metabolic bypass in living *E. coli* cells. In the triple deletion strain GL3, we compared the expression of the wild type gene (*fsaA*) with the allele encoding a variant enzyme (FSAA A129S) with a known enhanced catalytic efficiency for F6P as substrate. As a result, a glycolytic bypass leading to growth on glucose was only detected in GL3 with the plasmid-borne gene variant.

What differentiates the mutant *fsaA* gene from the wt *fsaA* gene? Apparently, the difference lies in the improved catalytic efficiency for F6P which is known for FSAA A129S [24,27,35]. FSAA has a rather high *K*_M_ value for F6P in the range of 7–19 mM as determined by various researchers (however not under the same conditions) ([24,27,35], and E Guitart Font, unpublished results). FSAA A129S however, showed an improved affinity for F6P (range from 1.5 to 4 mM) as well as an improved activity with this compound as substrate leading altogether to an approximate 20-fold improved catalytic efficiency in vitro ([24,27,35], and E Guitart Font, unpublished results).

If the affinity of the two FSA enzymes for F6P is important, knowledge on the intracellular F6P concentrations in *E. coli* could be helpful. The Rabinowitz group determined intracellular concentrations for F6P of about 2.5 mM and for G6P in the range of 8 mM for *E. coli* when grown on minimal medium with glucose as sole carbon source [66]. We attempted to estimate G6P and F6P concentrations in cells of LJ110 and GL3 strains. Cells were first grown on MM fructose, then washed and transferred to MM glucose and incubated for 2 h. Under these conditions, cells of wt LJ110 grew on glucose, while GL3 cells did not. G6P and F6P were then determined enzymatically after extraction from cells by a boiling method. Concentrations of F6P and G6P stayed below the threshold values in LJ110 (<10 µM for sugar phosphates). Cells of GL3 showed intracellular concentrations of G6P at 1.6 ± 0.6 mM and F6P of 1.1 ± 0.4 mM (average values of two biological replicates, each with three technical replicates). While we are aware that our conditions differed from that of [66] and that the concentrations of the two hexose phosphates may be determined with more accuracy, the results obtained in our study nonetheless allow the statement that the intracellular F6P concentration in GL3 is far from saturating for FSAA while FSAA A129S would be at least closer to its optimal rate.

Growth of GL3/pJF119*fsaA^A^*^129*S*^ on glucose was restored with a mean generation time of about 6 h (µ of 0.12 h^−1^) whereas the wt strain LJ110 in our study showed a µ of 0.59 h^−1^. Therefore, the FSAA A129S reaction is not a simple stand-in for the PFK function in the GL3 strain. Several limitations in central metabolism can be envisioned: insufficient enzyme activities and efficiencies, lack of NADPH redox cofactors, inappropriate effector provision and unwanted side effects of a new metabolite, to name a few. Here we will discuss some of these issues and questions.
(a)We have shown here that a bypass of the PFK block mutation is possible if a suitable enzyme activity of FSA is present in *E. coli* cells. Cleavage of F6P into G3P and DHA with subsequent phosphorylation of DHA would lead to the same couple of triose phosphates as in EMP, and no extra energy expenditure is necessary (see Figure 4). Therefore, one might ask why did Nature not choose this pathway as a general alternative to EMP? The reason could be that DHA may not be funneled completely or fast enough into DHAP. DHA is a short chain sugar which is well-known to readily react with proteins as it forms adducts with lysine and arginine residues [67] leading to browning; this effect is exploited for skin tanning [68], where DHA is an ingredient in many self-tanning compounds. We also have observed that during the course of incubation, only the cultures of the strains that produce DHA became yellow. Still, DHA can also lead to protein inactivation and it is known as a mutagen for *E. coli* under certain circumstances [67]. Both consequences would be detrimental for cell metabolism and genetic stability. This might be the reason why an elaborate high affinity PTS for DHA phosphorylation (DHAKLM) is present in *E. coli* [5,58,69] whose main function arguably lies not so much in catabolism (yielding DHAP which can be catabolized in glycolysis or used as building block for glycerophospholipid formation) but rather in effective detoxification of an undesired metabolite. We found increased gene activity of the *dhaK* gene in GL4 which we take for evidence that the DHAKLM system is indeed involved in further metabolism of DHA which stems from F6P cleavage.(b)FSAA A129S not only opens a bypass at the block between F6P and F1,6BP (see Figure 4 and Appendix A). Through its action, G3P is formed which can be further metabolized in the lower glycolytic trunk to PEP and ultimately pyruvate. As well, G3P together with F6P serves as substrate for transketolase (TKT) from the PPP. TKT transfers two-carbon dihydroxyethyl units from F6P to G3P forming E4P and X5P; transaldolase then performs a DHA transfer reaction from F6P onto E4P yielding G3P and S7P [70,71]. FSA could also add DHA on E4P to form S7P and be a sink for DHA [25]; thus, not all DHA is to be expected to go to glycerol (in strain GL7).(c)The block of PFK in glycolysis leads to accumulation of phosphorylated sugars (G6P, F6P) from PTS substrates and causes the so-called sugar phosphate stress. This elicits activation of SgrS [59,72,73]. *SgrS* sRNA interacts with *ptsG* mRNA to reduce translation of new PtsG (enzyme IIBC^Glc^) molecules [74] whereas already existing PtsG molecules are still active as protein turnover is slow [75]; *SgrS* sRNA also activates synthesis of a sugar phosphatase (YigL) which dephosphorylates G6P to help with glucose homeostasis [75]. This could lead to less G6P and, in turn, to less F6P in the cells. Moreover, the gene product of *sgrS*, when expressed ectopically, is a small polypeptide SgrT which inhibits PtsG activity in vivo [76,77]. Thus, although FSAA A129S removes F6P, the uptake rate of glucose might be lower in GL3 than in LJ110 and contribute to the slower growth on glucose.(d)The missing glucose 6-phosphate dehydrogenase function (Δ*zwf*) in strain GL3 and the resulting lack of a subsequent 6-phosphogluconate dehydrogenase step pose another problem as the NADPH supply by these two enzymes is not restored by introduction of FSAA A129S in the bypass pathway. Cells of GL3/pJF119*fsaA^A^*^129*S*^ have thus to rely either on transhydrogenase or isocitrate dehydrogenase as NADPH sources [45] for anabolism. This could also contribute to the observed slower growth rates.(e)Why does the triple mutant GL3 grow slower on C sources other than glucose? Fructose is in part transported via the enzyme II for D-mannose to yield F6P [14] and thus could end up at the same blockade as with glucose. Xylose and other C sources which are initially routed through the PPP, also deliver F6P which could then accumulate and/or be interchanged with G6P. Both would not contribute to the EMP. In comparison to GL3, in GL3/pJF119*fsaA^A^*^129*S*^ and strain GL7 growth on xylose and other C sources was only partially restored while being clearly slower than in the wild type, LJ110 (data not shown). This warrants further investigations.

Another problem in the bypass pathway lies in the provision or recycling of PEP. As DHAKLM depends on PEP for the phosphorylation step (such as in strain GL3/pJF119 *fsaA*^A129S^ or in GL4), 2 moles of PEP are necessary for activation of glucose (via the glucose PTS) and of DHA, while only 2 PEP are formed during the catabolism of glucose through the EMP (see Appendix A). Thus, in theory, no net PEP would remain to fulfill its many roles as substrate of pyruvate kinases (PYKA and PYKF, for ATP production), PEP carboxylase (PPC, anaplerotic reaction) [78], DAHP-synthases and EPSP synthase (in aromatic biosynthesis), or MurA (cell wall biosynthesis). The cell would thus have to rely on PEP recycling to come up with its various anabolic and anaplerotic needs. PEP could be recycled from pyruvate by a PEP synthetase (PPS), from pyruvate via malate (malic enzyme, pyruvate + HCO^3−^ + NAD(P)H_2_ > malate + NAD(P)^+^) in conjunction with malate dehydrogenase and PEP carboxykinase (PCK; oxaloacetate + ATP > PEP + ADP). In any way, this would afford ATP.

An additional problem which may contribute to slow growth of cells which have the bypass pathway is the low concentration of F1,6BP. This molecule is not only an intermediate of the EMP, it also plays a role as a “sensing metabolite” in *E. coli* [66,79] as it activates enzymes of the lower glycolytic trunk (PYKF) and in anaplerosis (PPC), and thereby contributes to faster glycolytic metabolism. As the FSA-based bypass does not directly lead to F1,6BP, however, the enhancement of PYKF and PPC might be less, which could explain the slower growth of GL3-derived strains on compounds via F6P. We do not know how much F1,6BP is formed from the gluconeogenic aldol addition condensation of G3P + DHAP.

Having shown that a plasmid-encoded *fsaA*^A129S^ restored growth on glucose, we went on to integrate a copy of the mutant gene under control of an inducible P_tac_ promoter in the chromosome of GL3 to come up with GL4. GL4 grew slower on glucose (µ 0.05 h^−1^) apparently due to the lower gene dosage. Transiently, DHA appeared in cultures on glucose but was consumed again. GL4 reached nearly the same cell yield as LJ110 (Table 4). By deletion of the *dhaKLM* genes, GL5 was constructed. This strain produced up to 3 mM of DHA but this still disappeared during further incubation. The fate of DHA here is less clear. It might be that DHA is first reduced to glycerol by GLDA and then phosphorylated by GLPK to glycerol 3-phosphate or it could be phosphorylated directly by GLPK which has a high activity but low affinity (*K*_M_ value of ≈0.5 mM) towards DHA [80]. Indeed, in strain GL6 which has the *glpK* gene deleted, DHA accumulated up to 8.8 mM. As a new product, glycerol appeared as minor by-product (up to 1.9 mM). Most likely DHA was in part converted to glycerol by GLDA which is known to act preferentially on DHA [81]. As GLPK is missing now, glycerol is a dead end product. To our best knowledge, formation of glycerol by *E. coli* cells from glucose via DHA is a novel feature (“new-to-nature pathway”).

This pathway was reached by expression or alteration of genes which are already present in *E. coli*. To be efficient, a single amino acid substitution (A129S in FSAA) and a change of promoters (P_tac_ instead of the native promoter of *fsaA*) were necessary. A comparison of the novel glycerol pathway with the “yeast glycerol pathway” made possible by introduction of DAR1/GPD1 and GPP genes from yeast in *E. coli* shows that the novel pathway (from F6P to glycerol) consumes 1 mol ATP less than the yeast pathway from F6P via F1,6BP to DHAP, glycerol 3-P and finally glycerol [41,42,43]. The novel glycerol pathway, however, requires NADH as reducing agent. In its present state, it has an upper theoretical limit of 1 mole glycerol/mole of glucose; the remaining C atoms of F6P flow into G3P and the lower glycolytic pathway or into the PPP.

As already stated above, PEP is needed for glucose uptake via the PTS. In our best glycerol producer (strain GL7), glycerol and DHA are end products and together almost make up half (≈42%) of the carbon which is generated from the fed glucose. The residual carbon can be used for generation of only approximately 1.2 moles of G3P (from 1 mole of glucose) and subsequently only 1.2 moles of PEP should be gained from glucose. On the other hand, 1 mole PEP is necessary as phosphoryl group donor for glucose uptake via the PTS. As discussed above, several enzymes involved in catabolic, anaplerotic, or anabolic reactions would compete for the PEP provided by the EMP. Therefore, it does not come as a surprise that generation times on glucose are much higher in GL7 (≈25 h) and the growth yield from this C source is only about 30% of that of wild type (Table 4).

The many challenges and imbalances (discussed above) which lead to the observed slow growth in strains GL5 and GL7, however, can on the other hand be seen as starting point of adaptive evolution. Through selection and analysis of faster growing descendants we want to find out which metabolic solutions will be found by these evolved strains (E Guitart Font, M Wolfer, GA Sprenger, manuscript in preparation).

## 4. Materials and Methods

### 4.1. Chemicals, Carbon Sources, and Enzymes

Antibiotics, fine chemicals, and sugars were purchased from Carl Roth GmbH (Karlsruhe, Germany), Gerbu Biotechnik GmbH (Heidelberg, Germany), Honeywell Specialty Chemicals Seelze GmbH (Seelze, Germany), Merck KGaA (Darmstadt, Germany) and Sigma-Aldrich Chemie GmbH (Taufkirchen, Germany). All chemicals were of the highest purity. Auxiliary enzymes were from Sigma-Aldrich. Restriction enzymes, polymerases, and T4 ligase were from New England Biolabs GmbH (Frankfurt am Main, Germany), Genaxxon bioscience GmbH (Ulm, Germany) and Roche Diagnostics GmbH (Mannheim, Germany). Oligonucleotides were synthesized by biomers.net (Hohenbrunn, Germany).

### 4.2. Bacterial Strains, Plasmid and Strain Constructions

Bacterial strains and plasmids of the present study are shown in Table 1 and Table 3. Molecular biological protocols followed general procedures [82] or specific manufacturer’s instructions. In the early state of the project, chromosomal gene deletions were performed according to a recombineering method [47]. Later on, deletions and integrations were carried out with the CRISPR-Cas9 method for *E. coli* [48]. Genes and chromosomal fragments were amplified by PCR from plasmid DNA and *E. coli* K-12 chromosomal DNA. Colony PCR and agarose gel electrophoresis were to verify the integrity of all constructed plasmids and chromosomal insertions/deletions. PCR primers are given in Appendix A; agarose gels are shown in Appendix A. Custom DNA sequencing was performed at GATC Biotech AG (Konstanz, Germany).

### 4.3. Growth Conditions

Bacteria were typically grown in Luria–Bertani (LB)-medium [82] at 30 or 37 °C (depending on the plasmid used) and at 180–200 rpm (INFORS HT, Bottmingen, Switzerland). If necessary, media were supplemented with antibiotics (100 µg/mL ampicillin, 50 µg/mL kanamycin, 50 µg/mL spectinomycin, or 25 µg/mL chloramphenicol).

For growth experiments, a minimal medium (MM) [50] was used and supplemented with 20 µg/mL thiamine and the corresponding single C-source. Precultures were inoculated from MM agar plates with 28 mM fructose as C source and incubated overnight in 5 mL liquid media at 37 °C and 200 rpm on MM containing 28 mM fructose and 100 µM IPTG. The next day, 2 mL of the preculture were centrifuged at 16,500× *g* for 1 min and washed twice with 1 mL MM containing the corresponding C-source (intended starting concentrations were 28 mM D-glucose; 33 mM D-xylose; 33 mM L-arabinose, or 28 mM D-galactose; note that the actual starting concentrations—after inoculation with cells—were determined separately by HPLC for measurements of C source consumption and product formation). After another centrifugation (16,500× *g* for 1 min), the pellet was resuspended in 1 mL MM with the corresponding C-source. The main culture (50 mL MM containing the corresponding C-source, 100 µM IPTG and, if necessary, antibiotic in a 500 mL shake flask with a metal cap) was inoculated to a starting OD600 nm of ≈0.05. The cultures were incubated at 200 rpm and 37 °C until the stationary phase was reached. During growth, OD600 nm was monitored and samples were withdrawn and centrifuged at 16,500× *g* for 5 min. The resulting supernatants were stored at −20 °C for HPLC analysis. At the stationary phase, cells were harvested by centrifugation (3400× *g*, 10 min at 4 °C), washed in 2 mL glycylglycine buffer (50 mM, pH 8.5), centrifuged again, and the resulting pellet was stored at −20 °C until enzyme activity assays.

Growth experiments with fructose as sole C-source were performed in a main culture with a total volume of 25 mL (250 mL shake flask with a metal cap). In this case, OD600 nm was followed until the stationary phase was reached.

### 4.4. Quantitative Real-Time PCR (qPCR)

RNA was extracted from cells grown on MM containing 28 mM glucose and 100 µM IPTG during the mid-exponential phase (OD600 nm 0.8–1.4) using the RNeasy Mini Kit (Qiagen GmbH, Hilden, Germany) with on-column DNA treatment. Total RNA was reverse-transcribed to cDNA with random primers using the RevertAid reverse transcriptase (Thermo Fisher Scientific Inc., Waltham, MA, USA). Total cDNA and specific primers (see Appendix A) were added to Biozym Blue S’Green qPCR mix (Biozym Scientific GmbH, Hessisch Oldendorf, Germany). qPCR reactions were performed in a qTower 2.0 (Analytik Jena AG, Jena, Germany) and results were analyzed with the software qPCRsoft 3.4 from the same manufacturer. *ftsZ* gene was used as the reference gene.

### 4.5. Preparation of Cell-Free Extracts (cfe) and Enzyme Activity Assays

Cells were disrupted by adding 200 U/mL lysozyme and 40 Kunitz U/mL DNase I (95 rpm, 30 min at 30 °C). The suspensions were then centrifuged (18,000× *g*, 1 h at 4 °C) and the resulting cell-free extract (cfe) was directly used for measurement of GLDA activity. For the determination of FSA activity, the cfe was treated by incubation at 75 °C for 20 min and centrifuged again. The resulting cell-free supernatants were used in the FSA activity assay. Both enzymatic activities were measured at 30 °C by monitoring the decrease in NADH concentration (0.3 mM) at 340 nm in a Cary 60 uv/vis spectrophotometer with the CaryWinUV version 5.0.0.999 software (Agilent Technologies, Waldbronn, Germany). The cleavage reaction of F6P (7.5 mM for FSAA A129S and 27 mM for FSAA) into DHA and G3P was performed as described earlier [20]. For the GLDA reaction, the reduction of DHA (1 mM) was carried out as reported by Subedi et al. [81].

### 4.6. HPLC Analysis of Sugars, DHA, and Glycerol in Supernatants

Concentrations of sugars, DHA, and glycerol were determined by HPLC analysis, performed on an HPLC instrument (1100 series) with a RID detector, both from Agilent Technologies (Waldbronn, Germany). An organic acid column, 300 × 8 mm from Chromatographie-Service GmbH (Langerwehe, Germany) was used. The mobile phase consisted of 5 mM H2SO4 with a flow rate of 0.6 mL min^−1^. Raw data were analyzed using the Agilent software “ChemStation for LC 3D Systems”.

## Figures and Tables

**Figure 1 ijms-21-09625-f001:**
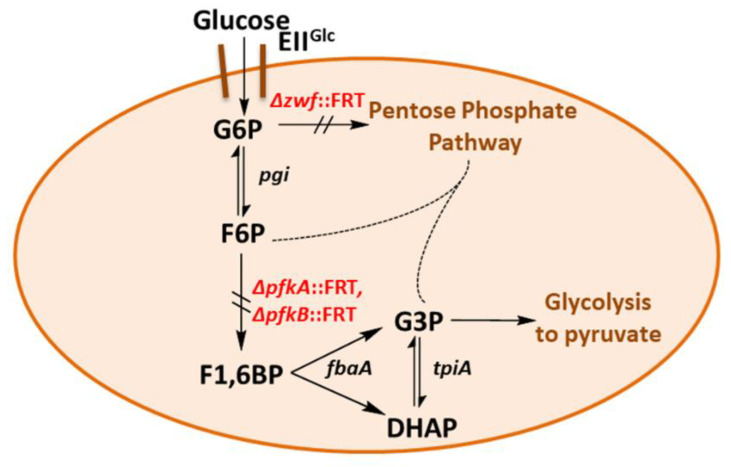
Schematic of the glycolytic blockage in the triple deletion strain, GL3. Genes encoding relevant enzyme activities are *fbaA* (fructose 1,6-bisphosphate aldolase A), *pfkA* (phosphofructokinase A), *pfkB* (phosphofructokinase B), *pgi* (phosphoglucose isomerase), *tpiA* (triosephosphate isomerase), *zwf* (glucose 6-phosphate dehydrogenase), EII^Glc^ (glucose-specific enzyme II of the PEP-dependent sugar:phosphotransferase system, PTS), FRT = FLP recognition site.

**Figure 2 ijms-21-09625-f002:**
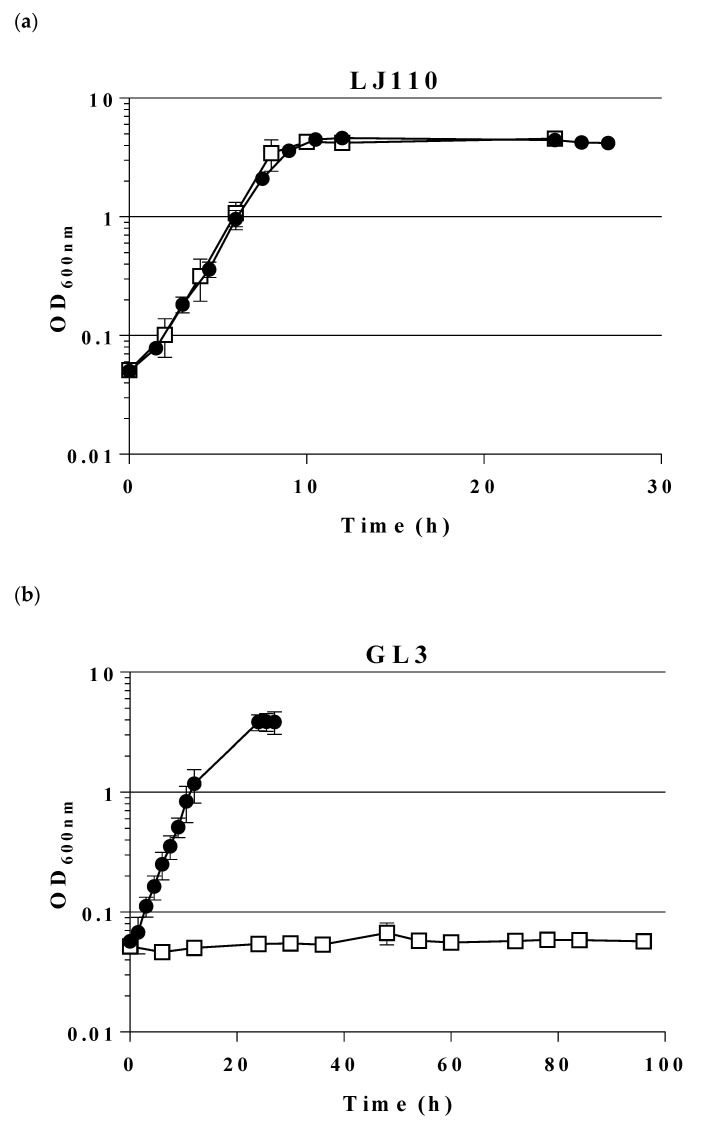
Growth behavior of wt LJ110 (**a**) and triple mutant strain, GL3 (**b**) on glucose or fructose as carbon sources. Shake flask growth on MM with fructose and 100 µM IPTG (●) or on MM with glucose and 100 µM IPTG (□). At 37 °C, 200 rpm. Average values of two biological replicates are presented. Both strains were inoculated from MM fructose overnight cultures. Note that the time axis is different in the two graphs. For details see Section 4.

**Figure 3 ijms-21-09625-f003:**
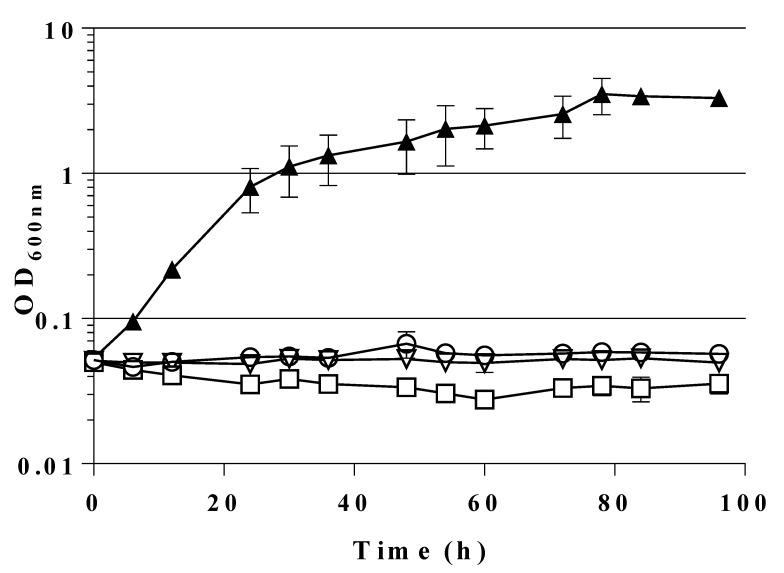
Growth behavior of GL3 (○), GL3/pJF119EH (▽), GL3/pJF119*fsaA* (□), and GL3/pJF119*fsaA*^A129S^ (▲). Shake flask cultures (37 °C) on MM with glucose as sole C source (100 µM IPTG, ampicillin 100 µg/mL if appropriate). Note that data for GL3 are taken from Figure 2 for comparison. Mean values from two independent biological replicates are presented. Precultures (overnight) on MM fructose + 100 µM IPTG were washed and inoculated to a starting OD_600_ of approx. 0.05. For details of growth measurements see Section 4.

**Figure 4 ijms-21-09625-f004:**
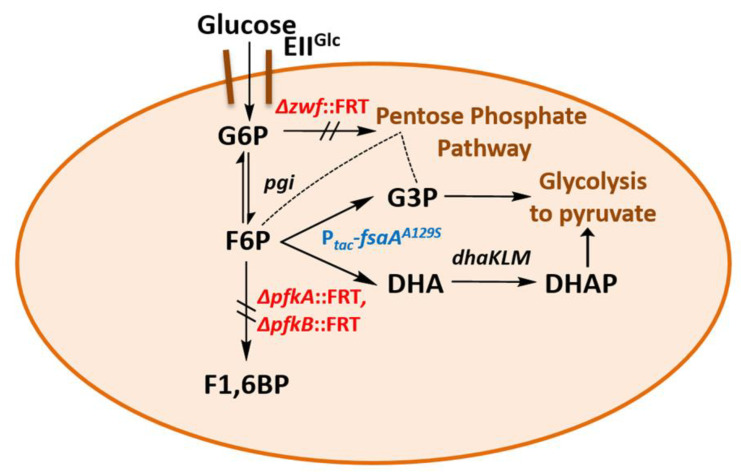
Schematic of a glucose bypass metabolism in the strains GL3/pJF*fsaA*^A129S^ and GL4. DHA, dihydroxyacetone; *dhaKLM*, genes for dihydroxyacetone kinase; DHAP, dihydroxyacetone phosphate; EII^Glc^, glucose transporter; F1,6BP, fructose 1,6-bisphosphate; F6P, fructose 6-phosphate; FRT, FLP recognition target; *fsaA*, fructose 6-phosphate aldolase gene; G3P, glyceraldehyde 3-phosphate; G6P, glucose 6-phosphate; *pfkA, pfkB*, phosphofructokinase genes; *pgi*, phosphoglucose isomerase gene; P_tac_, *tac* promoter; *zwf*, glucose 6-phosphate dehydrogenase gene.

**Figure 5 ijms-21-09625-f005:**
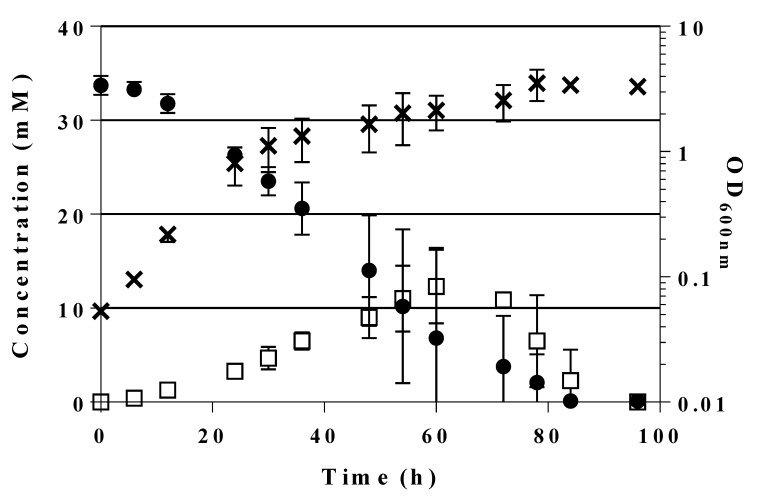
Glucose consumption (●) and DHA accumulation (□) in the culture supernatant of GL3/pJF119*fsaA*^A129S^ during growth on MM glucose and 100 µM IPTG over the time course of 96 h. OD_600 nm_ (✖). The actual concentrations of glucose and DHA were determined by HPLC. Mean values and standard deviations from two independent biological replicates and two technical replicates each.

**Figure 6 ijms-21-09625-f006:**
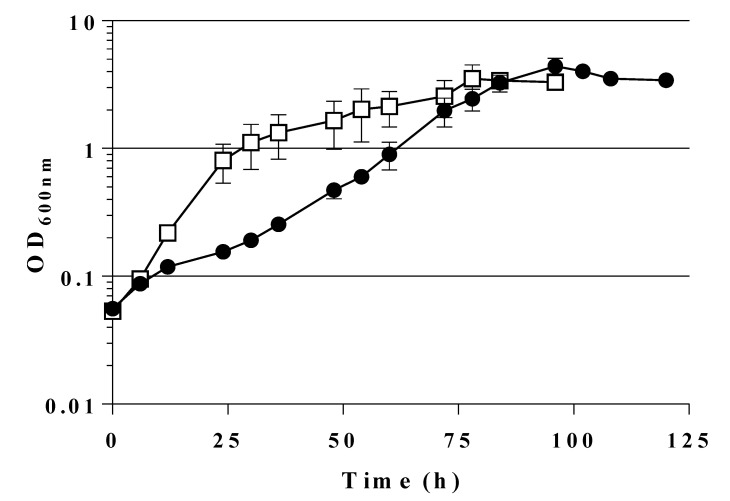
Growth behavior of GL3/pJF119*fsaA*^A129S^ (□) in comparison with GL4 (●). Shake flask growth on MM with glucose, 100 µM IPTG and, if appropriate, 100 µg/mL Amp. Precultures were on MM fructose and 100 µM IPTG to induce FSAA A129S activity already before the change of C source. Average values of two independent biological replicates are presented.

**Figure 7 ijms-21-09625-f007:**
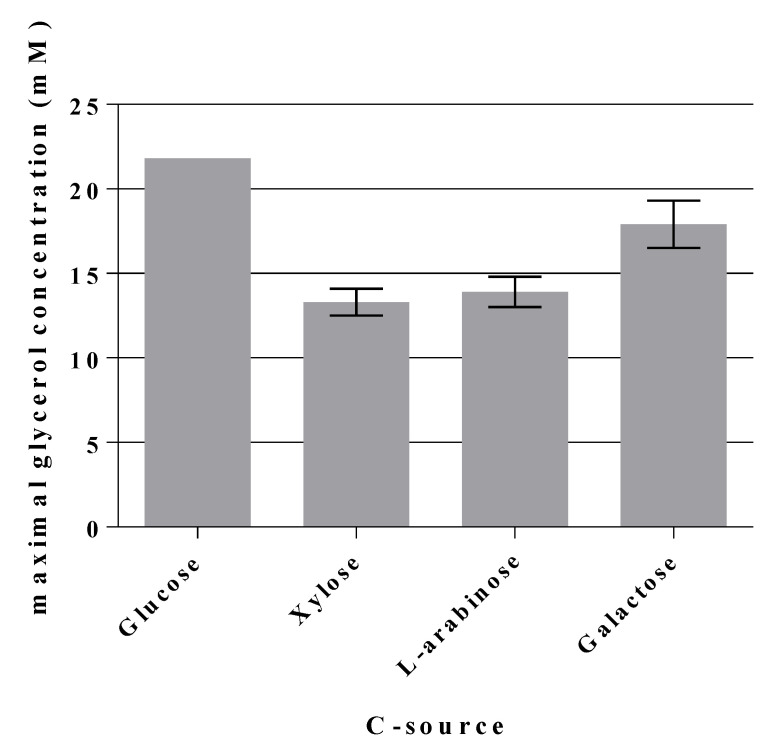
Maximal glycerol concentrations formed by GL7 during growth on different C-sources. Hexoses (D-glucose, D-galactose) were set at ≈28 mM initial concentrations, pentoses (L-arabinose, D-xylose) at ≈33 mM. Actual starting concentrations were determined by HPLC. Mean values of two independent biological replicates are given.

**Table 1 ijms-21-09625-t001:** List of bacterial strains used in this study. All strains, except DH5α, are derived from *Escherichia coli* K-12 W3110 strain (LJ110). Methods employed to create the strains: ^a^ Datsenko and Wanner [47] and ^b^ CRISPR/Cas [48].

Strain	Genotype	Origin/Reference
DH5α	*F^–^, Φ80d, lacZ*Δ*M15,* Δ*(lacZYA-argF) U169, recA1, endA1,* *hsdR17 (r_K_^–^, m_K_^+^), phoA, supE44, λ–, thi-1, gyrA96, relA1*	[49]
LJ110	W3110 *fnr**^+^*, wild type	[44]
GL1	LJ110, Δ*zwf*::FRT	This study ^a^
GL2	LJ110, Δ*zwf*::FRT Δ*pfkB*::FRT	This study ^a^
GL3	LJ110, Δ*zwf*::FRT Δ*pfkB*::FRT Δ*pfkA*::FRT	This study ^a^
GL4	LJ110, Δ*zwf*::FRT Δ*pfkB*::FRT Δ*pfkA*::FRT Δ*lacZ*::P_tac_-*fsaA*^A129S^	This study ^b^
GL5	LJ110, Δ*zwf*::FRT Δ*pfkB*::FRT Δ*pfkA*::FRT Δ*dhaKLM* Δ*lacZ*::P_tac_-*fsaA*^A129S^	This study ^b^
GL6	LJ110, Δ*zwf*::FRT Δ*pfkB*::FRT Δ*pfkA*::FRT Δ*dhaKLM* Δ*lacZ*::P_tac_-*fsaA*^A129S^ Δ*glpK*	This study ^b^
GL7	LJ110, Δ*zwf*::FRT Δ*pfkB*::FRT Δ*pfkA*::FRT Δ*dhaKLM* Δ*lacZ*::P_tac_-*fsaA*^A129S^ Δ*glpK* Δ*rbsk*::P_tac_-*gldA*	This study ^b^

**Table 2 ijms-21-09625-t002:** Growth phenotypes of wild type LJ110 and mutant strains GL3 and GL4 on agar plates. LB-agar; minimal medium (MM)-agar [50] with 0.5% C-source (*w*/*v*). normal = single colonies with a diameter of ≥1 mm; small = colonies with a diameter ≤1 mm; in grey = barely visible; - = no growth; * = occurrence of papillae after several days of incubation. ** Lactose-minus due to gene insertion (1) colonies evaluated after 1 day of incubation at 37 °C; (2) colonies evaluated after 2 days of incubation at 37 °C. Average of at least three independent biological replicates. For details see Section 4.

Strain:		LJ110	GL3	GL4
LB-agar		Normal (1)	Normal (1)	Normal (1)
MM-agar + 0.5% C-source (*w*/*v*)	No C-source	-	-	-
	D-Fructose	Normal (1)	Small (2)	Small (2) *
	D-Glucose	Normal (1)	-	Small (2)
	Glycerol	Normal (1)	Small (1)	Small (1)
	Lactate (pH 7.0)	Small (1)	Small (1)	Small (1)
	Mannitol	Normal (1)	-	-
	Succinate (pH 7.0)	Small (1)	Small (1)	Small (1)
	Sorbitol	Normal (1)	-	Small (2) *
	D-Xylose	Small (1)	-	Small (2)
	Maltose	Normal (1)	-	-
	D-Galactose	Small (1)	-	-
	Gluconate (pH 7.0)	Normal (1)	Small (2)	Small (1)
	L-Arabinose	Normal (1)	Small (2)	Small (2)
	Lactose	Normal (1)	-	- **

**Table 3 ijms-21-09625-t003:** Plasmids used during this study. RBS, ribosome binding site; ts, temperature-sensitive replication. Amp^R^, ampicillin resistance gene; Cm^R^, chloramphenicol resistance gene; Km^R^, kanamycin resistance gene; Spc^R^, spectinomycin resistance gene.

Plasmid	Relevant Characteristics	Source/Reference
pJF119EH	P_tac_, *lacI*^q^, RBS, Amp^R^	[51]
pJF119*fsaA*	*fsaA* wildtype gene, cloned into pJF119EH	[52]
pJF119*fsaA*^A129S^	*fsaA*^A129S^ gene, cloned into pJF119EH	[52]
pJF119ΔEP_tac-_*gldA*	P_tac_*gldA*, lacI^q^, ΔEcoRI, ΔNdeI, optimized RBS, Amp^R^	Stefan Riemer (2010, unpublished)
pKD46	repA101(ts), araC, P_araB_-ϒ-β-exo (λred recombinase), Amp^R^	[47]
pCO1-cat	FRT-cat-FRT, Amp^R^, Cm^R^	[53]
pCP20	FLP^+^, λ cl857^+^, λ p_R_ rep^ts^, Amp^R^, Cm^R^	[54]
pCas	*repA*101 (Ts), P_araB_-γ-β-exo (λred recombinase), P_cas_-*cas*9, *lacI*^q^, P_trc_-sgRNA-pMB1, Km^R^	AddGene [48]
pTarget with sgRNAs	With (T) or without (F) donor DNA	AddGene [48]
pTargetF-*dhaKLM*	pMB1, sgRNA-*dhaKLM*, Spc^R^	This study
pTargetF-Cm-*glpK*	pMB1, sgRNA-*glpK*, Cm^R^	This study
pTargetT-Cm-Δ*lacZ*::P_tac_-*fsaA*^A129S^	pMB1, sgRNA-*lacZ*, ΔlacZ::P_tac_-*fsaA*^A129S^, Cm^R^	This study
pTargetF-Cm-*rbsK*	pMB1, sgRNA-*rbsK*, Cm^R^	This study
pJNTN-m-L	P_tac_, *lacI*^q^, Km^R^	Natalie Trachtmann (unpublished)
pJNTN-m-L-*pfkA*	*pfkA* gene cloned into pJNTN-m-L	Natalie Trachtmann (unpublished)

**Table 4 ijms-21-09625-t004:** Consumption of different carbon sources and formation of DHA and glycerol in shake flask cultivations. All shake flask cultivations contained MM medium with 100µM IPTG. The concentrations of the various sugars were set approximately at the values in the left column. Note that, due to unavoidable technical limitations in media preparations, not all shake flasks had the exact same starting concentrations. We therefore measured the initial sugar concentrations immediately after inoculation with cells and then followed the decrease by HPLC measurements. For details of growth conditions and measurements see Section 4. n.a., not applicable; n.g., no growth; Gt (min) is the generation time (in min) determined during the logarithmic growth phase. Res *, residual DHA or glycerol, respectively, at the end of cultivation.

C Source	Strain	Time (h) until Max. OD_600_	Max. OD_600_	C Source (mM) Consumed	DHA		Glycerol		µ (h^−1^)	G_t_ (min)
					Max. Conc. (mM)	Res *	Max. Conc. (mM)	Res *		
28 mM glucose	LJ110	24	4.562 ± 0.003	33.8	0.0 ± 0.0	No	0.0 ± 0.0	No	0.59 ± 0.01	71 ± 1
	GL3	n.g.	n.g.	n.a	n.a.	n.a.	n.a.	n.a.	n.a.	n.a.
	GL4	96	4.399 ± 0.675	32.2	0.5 ± 0.0	No	0.0 ± 0.0	No	0.05 ± 0.00	808 ± 11
	GL5	216	2.410 ± 0.489	30.2	3.1 ± 0.3	No	0.0 ± 0.0	No	0.02 ± 0.00	2293 ± 85
	GL6	288	1.497 ± 0.033	30.6	8.8 ± 0.3	Yes	1.9 ± 0.2	Yes	0.03 ± 0.01	1655 ± 103
	GL7	240	1.395 ± 0.165	30.2	3.5 ± 0.3	Yes	21.8 ± 0.0	Yes	0.03 ± 0.00	1526 ± 49
33 mM xylose	LJ110	24	4.492 ± 0.011	32.7	0.0 ± 0.0	No	0.0 ± 0.0	No	0.55 ± 0.01	76 ± 1
	GL7	48	4.431 ± 0.187	32.9	3.2 ± 0.1	Yes	13.3 ± 0.8	Yes	0.12 ± 0.01	361 ± 8
33 mM L-arabinose	LJ110	24	1.991 ± 0.107	34.5	0.0 ± 0.0	No	0.0 ± 0.0	No	0.53 ± 0.01	79 ± 4
	GL7	48	2.024 ± 0.061	35.0	4.1 ± 0.7	Yes	13.9 ± 0.9	Yes	0.12 ± 0.01	360 ± 21
28 mM galactose	LJ110	28	4.133 ± 0.160	28.0	0.0 ± 0.0	No	0.0 ± 0.0	No	0.19 ± 0.01	222 ± 21
	GL7	120	1.264 ± 0.001	28.2	3.5 ± 1.3	Yes	17.9 ± 1.4	Yes	0.04 ± 0.01	1266 ± 143

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
