# Peer review of "Opening a Novel Biosynthetic Pathway to Dihydroxyacetone and Glycerol in Escherichia coli Mutants through Expression of a Gene Variant (fsaAA129S) for Fructose 6-Phosphate Aldolaseâ€"

_ijms, 2020, doi:10.3390/ijms21249625_

Round 1
Reviewer 1 Report
The manuscript ijms-1018854 focuses on characterization of the mutant gene for 4 fructose 6-phosphate aldolase in Escherichia coli, and its role the creation of a new biochemical pathway from glucose to glycerol.
The reported data are new and interesting. The manuscript is well organized, the methodological approach is sufficiently correct and adequate to the research. The discussion is sufficiently argued. I consider that this manuscript is appropriate for publication in International Journal of Molecular Sciences.
I have only a couple of suggestions.
1) Although the purpose of the research becomes clear by reading the manuscript, it would be advisable for the Authors to explain it better in the introduction and also in the abstract.
2) The Authors should explain why the growth of GL3 on fructose as carbon sources was analyzed just to less than 30h.
Author Response
Reviewer #1
The manuscript ijms-1018854 focuses on characterization of the mutant gene for 4 fructose 6-phosphate aldolase in Escherichia coli, and its role the creation of a new biochemical pathway from glucose to glycerol.
The reported data are new and interesting. The manuscript is well organized, the methodological approach is sufficiently correct and adequate to the research. The discussion is sufficiently argued. I consider that this manuscript is appropriate for publication in International Journal of Molecular Sciences.
We thank the reviewer for the support and her/his nice words.
I have only a couple of suggestions.
1) Although the purpose of the research becomes clear by reading the manuscript, it would be advisable for the Authors to explain it better in the introduction and also in the abstract.
We appreciate the reviewer´s advice. We have and have added this to the introduction (in part moved from results section), new line 86 onwards of the revised manuscript (without markups, new numbering of quotations) as follows:
„DHA is an ingredient in many self-tanning compounds and glycerol is a carbon source which has received much attention as it can be used for production of many added-value compounds (for references see [36]). In contrast to other microorganisms such as yeast [37], cells of E. coli are not known to form and excrete glycerol when growing on glucose [38,39]. By introduction of genes from yeast which encode enzymes for the conversion of DHAP to glycerol, however, E. coli glycerol producers had been constructed [40-43]; eventually this has led to recombinant E. coli cells forming 1,3-propanediol from glucose via the intermediate glycerol. 1,3-propanediol is now used as monomer for the industrial production of the biobased polymer polytrimethylene terephthalate (PTT) [36,41]. As we observed transient formation of DHA from glucose with the newly developed strains described here, we decided to aim for a novel pathway to DHA and glycerol from glucose in E. coli mutants. This pathway is completely independent of heterologous genes.”
We hope that this helps to make the purpose of our research clearer.
2) The Authors should explain why the growth of GL3 on fructose as carbon sources was analyzed just to less than 30h.
We thank the reviewer for this question. Fructose was used as a quasi-neutral carbon source for comparison. As it became apparent that growth on fructose had become stationary after 27h of incubation, both in the wild type (constant OD values from 10h to 27h) and in GL3 (constant OD values of ~ 3.8 after 24 and 27h), we discontinued this measurement as further insights from this control were not expected. The incubation of GL3 on glucose, however, was continued as we wanted to find out whether a longer incubation would show signs of growth. As this was not the case after 96h of incubation (no doubling of OD value), this experiment was stopped, too. As described in the text, incubation on MM agar plates with glucose as sole C source did not yield visible growth after even 3 weeks of incubation.
Reviewer 2 Report
Comments on manuscript
Opening a novel biosynthetic pathway to dihydroxyacetone and glycerol in Escherichia coli through expression of a mutant gene (fsaAA129S) for fructose 6-phosphate aldolase in a pfkA, pfkB, zwf triple deletion strain
provided by Font and Sprenger
In their study the authors gave life to a synthetic pathway, bypassing EMP glycolysis by fructose 6-phosphate aldolase activity. They analyse the generated strains and engineer DHA and glycerol production from glucose, by deleting DHA and glycerol utilization and overexpression of glycerol dehydrogenase.
The science in the manuscript is straight forward and highly recommended for publication.
However, I have some concerns which I will describe in particular in the following.
General comments / major points:
The paper provides a synthetic pathway from glucose to DHA and glycerol. The authors explore both of the options. While the approach aims on generating these bioproducts I am missing an introduction regarding bioproductions of these products including markets aso.
The authors use mostly glucose as a source of carbon throughout their manuscript. From a biotechnological point of view, this makes sense, but as a metabolic engineer, it might have been supportive to start with a non-PTS carbon source. The effect of slow growth on glucose might have appeared due to the stoichiometric production of pyruvate by PTS driven glucose phosphorylation. When growing on glucose E. coli doesn’t “know” that it needs to express PEP-synthase. This might limit PEP availability and hence be a cause for slower growth.
In many cases, synthetic pathways can be optimized by adaptive laboratory evolution, either in a chemostat or by serial dilutions in cultivation tubes. Mostly achieving better growth in a reasonable time. I am missing an attempt to do this.
A strong tool of synthetic metabolism is the use of isotope tracing experiments. E.g. using 13C-1-glucose with subsequent amino acid mass analysis would help to ensure leaking of DHA back to metabolism (e.g. GL3+fsaA(mut)) and exclude this in GL5-7. This, however, would be a plus on top of the deletion work done.
In my personal opinion, the manuscript is in many parts unnecessary long. Not each and every aspect needs to be discussed, e.g. the absence of PKT. I would prefer if some sections could be reduced to make the manuscript more readable.
PpsA, must be a major player in the pathway. I strongly suggest to delete the gene in order to see of GL3+fsaA(mut) is still able to grow. The inability to grow will proof the the stoichiometry of the pathway. GL5 ΔppsA could still grow when the pathway DHAàGlycerolàGly3PàDHAP is working.
As said, ppsA transcription, which in glucose normally is not needed, must be higher in the engineered strains. My suggestion here would be to do qPCR on ppsA to see if it might be limiting.
Minor comments:
Title: Rephrase title. It is unnecessarily long and not very appealing.
Abstract: The abstract could be written in a more positive way, e.g. line 19
39: Furthermore oxPPP depends on PFK activity. 3 G6P will make 2 F6P and 1 G3P
54: This strain was used here https://www.nature.com/articles/s41467-020-19564-5 and here https://www.sciencedirect.com/science/article/pii/S0092867419312309
Figure 1: Only one of the figures is necessary, it should be clear to everyone that the WT situation is without deletions.
185: For a synthetic pathway carrying close to 100 % flux this is not slow. It is a very good starting point for e.g. ALE. As said above I would generally like to raise the point of presenting results in a more positive way.
190: Was the fsaA-WT activity when expressed in the same strain in the same range
241: Why? Please state earlier in the paragraph the reason behind your experiments.
249: As said above, could you also look at ppsA transcription levels? For the reasons mentioned earlier. DHA phosphorylation also is PEP dependent.
313: Where does the rest go?
DHA can do Maillard reaction with amines. As suggested earlier, experiments with e.g. 13C-1-labeled glucose would help to find out. The label will end up in DHA. If it will go back to metabolism it would be found in some isotope traces of amino acids.
324: Has been reported before, see above.
336: TKT and TAL cannot be used for fructose utilization. Their combined activity even consumes GAP + F6P and carbon will end up in C5. There is no stoichiometric flux to metabolism in a ΔzwfΔpfk strain possible via this route.
421: E. coli should be able to use its transhydrogenase efficiently when oxPPP is deleted.
Author Response
Reviewer 2:
Moderate English changes required
|
Yes |
Can be improved |
Must be improved |
Not applicable |
|
|
Does the introduction provide sufficient background and include all relevant references? |
( ) |
(x) |
( ) |
( ) |
|
Is the research design appropriate? |
(x) |
( ) |
( ) |
( ) |
|
Are the methods adequately described? |
(x) |
( ) |
( ) |
( ) |
|
Are the results clearly presented? |
( ) |
(x) |
( ) |
( ) |
|
Are the conclusions supported by the results? |
(x) |
( ) |
( ) |
( ) |
Answer to reviewer´s comments
Comments and Suggestions for Authors
Comments on manuscript
Opening a novel biosynthetic pathway to dihydroxyacetone and glycerol in Escherichia coli through expression of a mutant gene (fsaAA129S) for fructose 6-phosphate aldolase in a pfkA, pfkB, zwf triple deletion strain
provided by Font and Sprenger
In their study the authors gave life to a synthetic pathway, bypassing EMP glycolysis by fructose 6-phosphate aldolase activity. They analyse the generated strains and engineer DHA and glycerol production from glucose, by deleting DHA and glycerol utilization and overexpression of glycerol dehydrogenase.
The science in the manuscript is straight forward and highly recommended for publication.
We thank the reviewer for his/her appreciation and nice words
However, I have some concerns which I will describe in particular in the following.
General comments / major points:
The paper provides a synthetic pathway from glucose to DHA and glycerol. The authors explore both of the options. While the approach aims on generating these bioproducts I am missing an introduction regarding bioproductions of these products including markets aso.
We agree with this reviewer and reviewer #1 and have accordingly introduced a short statement on bioproduction of glycerol already in the introduction by moving a part from the results section and additional wording.
„DHA is an ingredient in many self-tanning compounds and glycerol is a carbon source which has received much attention as it can be used for production of many added-value compounds (for references see [36]). In contrast to other microorganisms such as yeast [37], cells of E. coli are not known to form and excrete glycerol when growing on glucose [38,39]. By introduction of genes from yeast which encode enzymes for the conversion of DHAP to glycerol, however, E. coli glycerol producers had been constructed [40-43]; eventually this has led to recombinant E. coli cells forming 1,3-propanediol from glucose via the intermediate glycerol. 1,3-propanediol is now used as monomer for the industrial production of the biobased polymer polytrimethylene terephthalate (PTT) [36,41]. As we observed transient formation of DHA from glucose with the newly developed strains described here, we decided to aim for a novel pathway to DHA and glycerol from glucose in E. coli mutants. This pathway is completely independent of heterologous genes.”
The authors use mostly glucose as a source of carbon throughout their manuscript. From a biotechnological point of view, this makes sense, but as a metabolic engineer, it might have been supportive to start with a non-PTS carbon source.
We thank the reviewer for this comment. Glucose is arguably still the favorite defined carbon source in biotechnology. The intended block in PFK and ZWF activities was therefore intentionally set so that F6P could accumulate from glucose and the role of FSA could be studied. This resulted in the discovery of an EMP bypass by the FSA variant. We could show that by further intentional genetic modifications, a novel pathway from glucose to glycerol could be established. This could thus serve as new paradigm pathway which can be studied further also by other researchers (most of them working still with glucose) as well.
The bypass and glycerol pathways we found now also allows (improved) growth of mutant strains with the triple deletion on non-PTS substrates such as galactose or the pentoses, D-xylose and L-arabinose as shown in table 4. We ourselves are not experts in unconventional C sources (from e.g. wood or straw hydrolysates ) but it is apparent that when the constituents D-galactose (could stem from whey lactose), D-xylose or L-arabinose (both constituents from hemicellulose) are used that natural mixtures of these might as well serve as source for glycerol production.
The effect of slow growth on glucose might have appeared due to the stoichiometric production of pyruvate by PTS driven glucose phosphorylation. When growing on glucose E. coli doesn’t “know” that it needs to express PEP-synthase. This might limit PEP availability and hence be a cause for slower growth.
We agree with the reviewer that the slow growth on glucose may (at least in part) be due to the PEP limitation exerted by the uptake system, PTS. We have discussed the need for PEP recycling in the discussion part. As outlined therein, the cells may respond to PEP limitation by enhancing other enzyme activities, one of them is PEP synthetase. The others would be malic enzyme in combination with malate dehydrogenase and PEP carboxykinase as described in the discussion.
In many cases, synthetic pathways can be optimized by adaptive laboratory evolution, either in a chemostat or by serial dilutions in cultivation tubes. Mostly achieving better growth in a reasonable time. I am missing an attempt to do this.
We agree with the reviewer that adaptive laboratory evolution (ALE) is a powerful tool for pathway optimization. With due respect, however, this was not the main focus of the present manuscript. [ this manuscript is intended as a contribution for a special collection on Microbial C-C bonding enzymes of IJMS].
We have pointed to ALE as outlook in the last sentence of our discussion. We regard the present manuscript as a first successful step towards the goal of including a FSA variant in pathway construction. For further optimization of the now opened pathway,we have, indeed, started with selection of faster growing clones by different approaches (e.g. serial dilutions as suggested by the reviewer, or from agar plates where larger colonies appeared on different C sources). Various faster growing clones (obtained from selections on various C sources such as glucose, galactose aso) are currently under investigation and intend to report a comparative analysis of these strains in due time. We have thus introduced this last sentence to the discussion:
„Through selection and analysis of faster growing descendants we want to find out which metabolic solutions will be found by these evolved strains (E Guitart Font, M Wolfer, GA Sprenger, manuscript in preparation).“
A strong tool of synthetic metabolism is the use of isotope tracing experiments. E.g. using 13C-1-glucose with subsequent amino acid mass analysis would help to ensure leaking of DHA back to metabolism (e.g. GL3+fsaA(mut)) and exclude this in GL5-7. This, however, would be a plus on top of the deletion work done.
We agree with the reviewer that isotope tracing experiments are a strong tool in synthetic metabolism research. We have described that we did not see a full stoichiometric formation of DHA and/or glycerol with our engineered strains. We discussed that a portion of DHA might be shuffled by FSA to Sed7P and thus into the PPP for biomass.
We agree that labeled glucose could help to follow the path of DHA in the cell and that this would be a plus on top of the work we showed. This is a difficult task, however, and as both authors are not experts in such an analysis, we refrained from this endeavor for the time being. [By the way, 13C label in the C1 position of glucose would end up 50:50 in the C1 and C3 carbons of DHA however, as DHA is a prochiral compound. If DHA is then phosphorylated by DHAKLM, the label would appear in a 50:50 ratio in the C1 and C3 atoms of DHAP. This might complicate the isotope analysis of amino acids].
In my personal opinion, the manuscript is in many parts unnecessary long. Not each and every aspect needs to be discussed, e.g. the absence of PKT. I would prefer if some sections could be reduced to make the manuscript more readable.
We thank the reviewer for his suggestion. We have now eliminated the part on phosphoketolase and on transketolase/transaldolase (lines 360-365 of old manuscript) as well as lines 378 to 381. We hope that this finds the support of the reviewers.
PpsA, must be a major player in the pathway. I strongly suggest to delete the gene in order to see of GL3+fsaA(mut) is still able to grow. The inability to grow will proof the the stoichiometry of the pathway. GL5 ΔppsA could still grow when the pathway DHAàGlycerolàGly3PàDHAP is working.
We thank the reviewer for this comment. Indeed, as already discussed above, PpsA may be a major player in PEP recycling. If we understand it correctly, the reviewer suggests to study whether DHA is phosphorylated by a PEP-dependent kinase (e.g. DHAKLM) or by an ATP-dependent glycerol kinase (either directly or via glycerol as intermediate)?. The difference would lie in PEP consumption. We have discussed the two possibilities in the text but we do not think that this is a major question and is a bit out of focus of the present report. According to our transcript measurements described in the text, we found relatively low (if at all) gene expression of gldA and glpK genes whereas dhaK was significantly increased. This is in favor of an assumption that DHAKLM is the main phosphorylating enzyme.
The ppsA deletion could be introduced in the said strains under normal working conditions in due time (although not in the 10 days we had to respond with a revised manuscript by IJMS). We are afraid, however, that under the prevailing pandemia with partial lockdown conditions which hamper work in the molecular biology labs at the moment, we will not come up with this task in the very near future. We will keep the suggestion in our minds, however, and will perform it with the next report (ongoing ALE as discussed above). With all due respect, however, we cannot perform this genetic work now.
As said, ppsA transcription, which in glucose normally is not needed, must be higher in the engineered strains. My suggestion here would be to do qPCR on ppsA to see if it might be limiting.
We agree with the reviewer that transcript measurements for ppsA would be of interest. They were not considered so when we chose the candidates for qPCR however. With regard to prevailing problems and inability of experimental work (see answer above) we respectfully refrain from repeating the transcript measurements at this time.
Minor comments:
Title: Rephrase title. It is unnecessarily long and not very appealing.
We have shortened and rephrased the title as follows:
„Opening a novel biosynthetic pathway to dihydroxyacetone and glycerol in Escherichia coli mutants through expression of a gene variant (fsaAA129S) for fructose 6-phosphate aldolase”
We hope that this finds the acclaim of the reviewer
Abstract: The abstract could be written in a more positive way, e.g. line 19
Thank you for this advice, we have accordingly changed line 19:
„allowed growth on glucose with a µ of about 0.12 h-1. A GL3 derivative with a chromosomally integrated copy of fsaAA129S (GL4) grew with a µ of 0.05 h-1 on glucose”
39: Furthermore oxPPP depends on PFK activity. 3 G6P will make 2 F6P and 1 G3P
We thank the reviewer for this advice and have thus inserted the following sentence (line 38):
„Furthermore, the oxidative pentose phosphate pathway (PPP) also depends on PFK activity”
54: This strain was used here https://www.nature.com/articles/s41467-020-19564-5 and here https://www.sciencedirect.com/science/article/pii/S0092867419312309
We thank the reviewer for this advice and we apologize for not having found these references on our own (one report however was published in Nature Communications the day after we submitted our manuscript). We have thus inserted three quotations for triple-negative strains described by various groups. However, none of these discussed growth on glucose or other C sources as we show here.
We have inserted the following sentence (line 54 of ms without markups): „Recently, three groups have reported the construction of E. coli triple deletion strains (DpfkA,DpfkB, Dzwf) but did not describe growth behavior on glucose [16-18]. [….] Such a triple-negative mutant would be expected to be devoid of growth on glucose and several other C sources (Figure 1).”
Figure 1: Only one of the figures is necessary, it should be clear to everyone that the WT situation is without deletions.
Following the advice of this reviewer, we have removed Fig1A and have changed the legend to Fig.1 accordingly.
185: For a synthetic pathway carrying close to 100 % flux this is not slow. It is a very good starting point for e.g. ALE. As said above I would generally like to raise the point of presenting results in a more positive way.
We thank the reviewer and to put the result in a more positive way, we have deleted „slowly“ here and at other occurrences in the text.
190: Was the fsaA-WT activity when expressed in the same strain in the same range
As we wrote in the manuscript, (line 181 of revised ms w/o markups) the wt FSA activity in strain GL3 was measured from cfe derived from fructose-grown cells only, as growth on glucose was not observed. The measured wt activity in heat-treated was 0.7 U/mg as described in the text. Compared with FSAA A129S (4.9 U/mg, after glucose growth) this was significantly less as described in the text (line 199). We interpret this as the lower catalytic efficiency of wt FSA. We think that we discussed this.
241: Why? Please state earlier in the paragraph the reason behind your experiments.
We have now inserted the following explanation (lines 235 „Our efforts to improve the formation of DHA was further studied are shown below.” And lines 253-255: “As we aimed for a better DHA formation, we were interested to see how genes of the glycerol and DHA metabolism were expressed in strain GL4; as well we looked for gene activities such as sgrS which we expected to be changed (see below)”
249: As said above, could you also look at ppsA transcription levels? For the reasons mentioned earlier. DHA phosphorylation also is PEP dependent.
As already lined out above as response to a similar question on PPS, „We agree with the reviewer that transcript measurements for ppsA would be of interest. They were not considered so when we chose the candidates for qPCR however. With regard to prevailing problems and inability of experimental work (see answer above) we respectfully refrain from repeating the transcript measurements at this time“
313: Where does the rest go?
DHA can do Maillard reaction with amines. As suggested earlier, experiments with e.g. 13C-1-labeled glucose would help to find out. The label will end up in DHA. If it will go back to metabolism it would be found in some isotope traces of amino acids.
We thank the reviewer for this question. We do not have a definitive answer yet. So we have included a new sentence at the end of the results section line 325: „Please note that the molar yield on glycerol/DHA is less than theoretically possible. For possible reasons see the discussion part.” And in the discussion we state that a portion of DHA might end up in the PPP via Sed7P as product of a second activity of FSA A129S on E4P.
324: Has been reported before, see above.
Please see our response above: „Recently, three groups have reported the construction of E. coli triple deletion strains (DpfkA, DpfkB, Dzwf) but did not describe growth behavior on glucose [quotations] […]. Such a triple-negative mutant would be expected to be devoid of growth on glucose and several other C sources (Figure 1).”
336: TKT and TAL cannot be used for fructose utilization. Their combined activity even consumes GAP + F6P and carbon will end up in C5. There is no stoichiometric flux to metabolism in a ΔzwfΔpfk strain possible via this route.
We thank the reviewer for this advice. We have thus deleted the sentence on TKT/TAL, together with the sentence on phosphoketolase.
421: E. coli should be able to use its transhydrogenase efficiently when oxPPP is deleted.
We agree with the reviewer on the role of transhydrogenase. However, we have acknowledged this role already in our discussion lines 447-448: „Cells of GL3/pJF119fsaAA129S have thus to rely either on transhydrogenase or isocitrate dehydrogenase as NADPH sources [quotation] for anabolism”
Submission Date
15 November 2020
Date of this review
25 Nov 2020 20:59:44
We thank the reviewer for his criticism and his positive advices. We hope that we could answer his questions in a satisfactory way.
Round 2
Reviewer 2 Report
The authors addressed my points to my satisfaction.